# Circulating hs-CRP, IL-18, Chemerin, Leptin, and Adiponectin Levels Reflect Cardiometabolic Dysfunction in Adults with Excess Weight

**DOI:** 10.3390/ijms26031176

**Published:** 2025-01-29

**Authors:** Óscar Javier Lara-Guzmán, Ángela María Arango-González, Rafael Álvarez-Quintero, Juan S. Escobar, Katalina Muñoz-Durango, Jelver Alexander Sierra

**Affiliations:** 1Vidarium–Nutrition, Health, and Wellness Research Center, Grupo Empresarial Nutresa, Carrera 52 #2-38, Medellin 050023, Colombia; ojlara@serviciosnutresa.com (Ó.J.L.-G.); amarango@noel.com.co (Á.M.A.-G.); jsescobar@serviciosnutresa.com (J.S.E.); kmunoz@serviciosnutresa.com (K.M.-D.); 2Grupo de Investigación en Ciencias Farmacéuticas-ICIF-CES, Facultad de Ciencias y Biotecnología, Universidad CES, Calle 10A #22-04, Medellin 050021, Colombia; ralvarezq@ces.edu.co

**Keywords:** cardiometabolic risk, obesity phenotypes, adipokines, biomarkers

## Abstract

Up to 30% of individuals with obesity may exhibit normal insulin sensitivity, a favorable lipid profile, and no signs of hypertension. This prompts the exploration of factors distinguishing cardiometabolically healthy individuals from those developing complications. This cross-sectional study included 116 individuals categorized into four groups by combining abdominal obesity and cardiometabolic health statuses. We compared circulating adipokines and gut microbiota composition between these groups. Individuals with abdominal obesity had higher levels of hs-CRP, TNF-α, MCP-1, IL-18, chemerin, and leptin, and a less favorable gut microbiota composition, including higher levels of potentially harmful bacteria (CAG-Pathogen) and lower levels of beneficial bacteria (CAG-Ruminococcaceae and CAG-Akkermansia), compared to those with adequate waist circumference. Those with obesity but cardiometabolically healthy displayed similar adipokine levels and microbiota composition to those with adequate waist. In contrast, individuals with abdominal obesity cardiometabolically abnormal exhibited significantly higher levels of hs-CRP, IL-18, chemerin, and leptin, and lower levels of adiponectin and CAG-Ruminococcaceae compared to those with abdominal obesity cardiometabolically healthy and adequate waist. Additionally, they differed in hs-CRP and adiponectin/leptin ratio from individuals with obesity cardiometabolically healthy. These findings suggest that altered adipokine profiles and gut microbiota may contribute to the development or persistence of cardiometabolic complications in obesity.

## 1. Introduction

Cardiovascular disease (CVD) and type 2 diabetes mellitus (T2DM) are major global health challenges, leading to significant morbidity and mortality worldwide. The economic and social burden of these diseases is substantial, impacting individuals, families, and healthcare systems. An estimated 20.5 million people died from CVDs in 2021, representing one-third of all global deaths [1]. Similarly, in 2022 the number of adults living with diabetes worldwide has surpassed 800 million [2] and is a leading cause of kidney failure, lower limb amputations, and blindness. The co-occurrence of obesity, T2DM, and CVD has become a significant public health concern, highlighting the need for a better understanding of the underlying mechanisms. The prevalence of obesity has increased in recent years, reaching pandemic levels [3]. The excessive accumulation of body fat is a major contributor to serious health problems. Elevated adiposity contributes to cardiometabolic abnormalities, such as insulin resistance and impaired glucose tolerance, hypertension, and dyslipidemia [4], which are independent risk factors that increase cardiovascular morbidity and mortality [5]. Without consensus, some authors categorize individuals with obesity and more than two cardiometabolic abnormalities as having cardiometabolically abnormal obesity [6]. On the other hand, individuals with excess weight with no additional cardiometabolic abnormalities, are described as having obesity cardiometabolically healthy [7,8]. Nevertheless, it has been argued that obesity cardiometabolically healthy is not a stable condition but a transient state at high risk of shifting toward the cardiometabolically abnormal phenotype over time [9,10,11].

Besides the association between adiposity and cardiometabolic risk, the adipose tissue plays a fundamental role in energy storage and is also an endocrine organ that produces various factors to regulate food intake, energy, and glucose homeostasis, called adipokines [12]. Adipokines encompass metabolic mediators (e.g., adiponectin, leptin, and serpin [13,14,15], acute-phase reactants (e.g., serum amyloid A, plasminogen activator inhibitor 1 [16], proinflammatory (e.g., chemerin, leptin, resistin, visfatin, lipocalin-2, osteopontin, WNT5A [16,17,18,19], and anti-inflammatory (e.g., adiponectin, SRFP5, omentin, and ghrelin [16]) molecules. Additionally, abdominal obesity and related cardiometabolic risk factors linked to an unhealthy adipose tissue expansion are accompanied by low-grade, chronic, systemic inflammation, linked to overexpressing proinflammatory factors, including chemokines (e.g., MCP-1, MIP-2, CCL5, IL-8, IFN-γ), damage-associated molecular pattern molecules, and cytokines, such as TNF-α, IL-6, and the interleukin-1 family (i.e., IL-1β, IL-18, IL-33, and IL-37) [16,20]. Among those, IL-18 and chemerin are primary inflammatory mediators in adipose tissue, emerging as promising biomarkers of obesity and cardiometabolic abnormalities with contrasting roles [21,22,23]. IL-18 is a potent proinflammatory cytokine that plays a crucial role in immune responses and inflammation [21,24]. Chemerin, on the other hand, is a chemoattractant that can promote the migration of immune cells to sites of inflammation. It is generally considered a proinflammatory mediator but has also been shown to have anti-inflammatory effects.

Given the established role of adipokines in the homeostasis of energy intake and insulin sensitivity, we hypothesize that the adipokine profile found in excess weight may offer complementary information to an adequate definition of cardiometabolic health status. By examining the specific patterns of adipokine secretion, we aimed to identify the underlying factors and changes that may predispose, delay, or protect individuals with excess weight from developing additional cardiometabolic abnormalities. We leveraged previous knowledge of the studied population and combined adipokine secretion with past measurements of additional emerging cardiometabolic risk factors, specifically the gut microbiota [25,26] and metabolites like trimethylamine (TMA) and trimethylamine N-oxide (TMAO), compounds that have been implicated in various health conditions, including cardiovascular disease, obesity, and metabolic disorders [27].

## 2. Results

The 116 participants of this study were adults of middle age, roughly distributed in equal proportions between males and females. Forty-four participants were classified as cardiometabolically healthy (37.9%) and 72 as cardiometabolically abnormal (62.1%) in health status.

To explore the relationships between central obesity and cardiometabolic health, subjects were classified based on waist circumference, a variable that accurately reflects body fat, particularly abdominal fat, and several cardiometabolic risk factors. As such, participants were classified into four phenotypes: lean healthy (adequate waist circumference, cardiometabolically healthy: 22.4% of the study population), lean abnormal (adequate waist circumference, cardiometabolically abnormal: 12.9%), obesity healthy (abdominal obesity, cardiometabolically healthy: 15.5%), and obesity abnormal (abdominal obesity, cardiometabolically abnormal: 49.1%). It is important to note that adequate waist circumference classification may include individuals classified as having adequate weight as well as individuals who are overweight based on BMI. This distinction acknowledges that excess fat can be stored in various regions of the body, such as the abdominal and gluteofemoral areas, which may not always correspond directly with BMI classifications. Z-scores allowed us to visually examine and identify patterns in the distribution of various health variables across these groups. Figure 1a,b show the observed heterogeneity within the four phenotypes. As a result of the classification, we observed a progressive trend in the health-associated parameters: the lean healthy and obesity abnormal were positioned as extremes, whereas the lean abnormal and healthy obesity had intermediate values. Blood pressure increased progressively from lean healthy to obesity healthy to lean abnormal to obesity abnormal. HDL cholesterol levels were lower in the metabolically abnormal health phenotype compared to the metabolically healthy. Conversely, VLDL and triglycerides were higher in the metabolically abnormal. LDL and total cholesterol were relatively similar across all four phenotypes. Fasting insulin, glucose, and HOMA indices indicated poor glycemic control in all groups except the lean healthy. Regarding the gut microbiota, there were important variations in this microbial community between the cardiometabolically abnormal and the cardiometabolically healthy, with the abundance of potentially pathogenic bacteria (CAG-Pathogen) rising more than five times in the cardiometabolically abnormal, independent of waist circumference. Finally, in the obesity abnormal, most adipokines were slightly increased, except for adiponectin, which was considerably lowered (Figure 1b).

The quantitative data for individuals classified into the four phenotypes are summarized in Table 1. First, the lean healthy phenotype had the most favorable cardiometabolic profile, a gut microbiota with the lowest abundance of potentially pathogenic bacteria (CAG-Pathogen) and the highest of beneficial bacteria (i.e., CAG-Akkermansia and CAG-Ruminococcaceae). It also had the lowest systemic inflammation, according to hs-CRP levels, and a favorable adipokine profile: the highest levels of adiponectin, the lowest levels of leptin, and the highest adiponectin/leptin ratio. In contrast, the lean abnormal phenotype, despite exhibiting low waist circumference, exhibited early signs of metabolic abnormalities, along with a lower adiponectin/leptin ratio, and higher systemic inflammation.

The two phenotypes with abdominal obesity had significantly higher adiposity compared to the lean healthy. The obesity healthy showed a milder metabolic profile than the obesity abnormal, with normal or slightly altered blood pressure and blood lipids. Glycemic control—with some borderline values—displayed significantly increased systemic inflammation and a distinct adipokine profile. The leptin and adiponectin levels were higher and lower, respectively, compared to the lean healthy phenotype. Yet, these values were intermediate to those observed in the obesity abnormal. The same trend toward increased levels of chemerin, MCP1, and IL-18, was observed in the obesity abnormal.

The phenotype displaying both abdominal obesity and established cardiometabolic abnormalities (i.e., obesity abnormal) showed the most concerning results. Compared to the obesity healthy, individuals had metabolic abnormalities that became more pronounced, highlighting the potential transition from a milder state to overt metabolic dysfunction. Their blood pressure was significantly higher, and their blood lipid profiles were markedly unhealthy, characterized by elevated circulating levels of VLDL, triglycerides, and lower levels of HDL cholesterol. While fasting blood glucose, HbA1c, and insulin levels fell within the normal range, they were significantly higher compared to other groups. Metabolic deterioration coincided with the lowest levels of adiponectin and the highest levels of leptin, a pattern strongly associated with adipose tissue dysfunction. Similarly to the lean abnormal phenotype, subjects with obesity abnormal had a significantly higher abundance of potentially harmful bacteria in their gut, accompanied by elevated hs-CRP levels, indicative of gut dysbiosis and systemic inflammation, which may have spilled over into the adipose tissue, as evidenced by the highest plasma levels of leptin, chemerin, IL-18, and MCP-1. All in all, the metabolic deterioration from obesity healthy to obesity abnormal correlated with altered levels of hs-CRP, leptin, adiponectin, chemerin, and IL-18.

The gut microbiota of individuals with abdominal obesity exhibited a lower abundance of potentially beneficial microorganisms belonging to CAG Ruminococcaceae and Akkermansia, and an increased abundance of bacteria from CAG-Pathogen (Appendix A). Similarly, individuals classified as obese by BMI had abundant bacteria belonging to CAG-Pathogen in the gut, whereas subjects with adequate weight had bacteria from CAG-Ruminococcaceae (Appendix A). After adjustment for age, gender, smoking status, and the city of origin, the inflammatory markers hs-CRP, IL-18, chemerin, leptin, and adiponectin, were strongly associated with changes in BMI and waist circumference (*p* < 0.05, MLR). It is noteworthy that when subjects were categorized based on gut microbiota (i.e., according to the dominant CAG), IL-18 was elevated in individuals with a dominance of the CAG-Pathogen (*p* = 0.025, MLR) (Appendix A), an association also observed in individuals with high BMI (Appendix A), and those with abdominal obesity that were cardiometabolically abnormal (Appendix A).

Consistent with previous findings, our analysis revealed similar trends when subjects were categorized independently by waist circumference, BMI, and cardiometabolic health status. Individuals with a BMI > 30 kg/m^2^, abdominal obesity, or cardiometabolic abnormal status demonstrated a higher prevalence of hypertension, impaired blood glucose control, adipose tissue dysfunction, and significant low-grade inflammation compared to those with a healthy status (Appendix A).

To understand the complex interplay between metabolic dysfunction and inflammation, we classified participants’ blood pressure, HDL, triglycerides, and insulin resistance (HOMA-IR) as healthy or abnormal, separately. Subsequently, we investigated the levels of hs-CRP, IL-18, chemerin, leptin, and adiponectin in each group. A key finding was the significant reduction in plasma adiponectin levels and elevated hs-CRP levels, observed in all groups with any type of metabolic abnormalities compared to healthy individuals (Figure 2). In contrast, the levels of IL-18 and chemerin, although elevated in subjects with all metabolic abnormalities, only showed statistically significant increases in participants with specific abnormalities, namely high blood pressure, high triglycerides, and low HDL cholesterol (*q* < 0.05; Figure 2a–c). On the other hand, leptin displayed a contrasting pattern in which high levels were observed in individuals with insulin resistance or low HDL (*p* < 0.001; Figure 2b,d), while subjects with high blood pressure and abnormal levels of triglycerides and had the lowest levels of this adipokine (Figure 2a,c).

We found strong correlations between hs-CRP, adiponectin, chemerin, and IL-18 with risk factors for cardiometabolic syndrome, including abdominal obesity, hypertension, low HDL, and poor overall cardiometabolic health. Chemerin showed a positive association with IL-18 and a negative association with HDL and adiponectin. IL-18 was positively associated with the abundance of pathogenic bacteria in the gut microbiota (CAG-Pathogen), hs-CRP, and chemerin, but negatively associated with adiponectin (Appendix A).

## 3. Discussion

Adipose tissue is not simply a reservoir of fat but rather constitutes an active endocrine organ with multiple functions. One of these roles is the participation in the inflammatory process in vascular and non-vascular tissues. Activated macrophages, along with adipocytes, secrete protein factors called adipokines that display metabolic, pro- and anti-inflammatory properties [28]. Excessive fat accumulation in the body could lead to adipose tissue dysfunction, resulting in dysregulated adipokine production, which contributes to the pathogenesis of cardiometabolic complications. Conventional clinical tools lack the sensitivity to detect these subtle, yet crucial indicators of future cardiometabolic risk. Thus, identifying early signs of adipose tissue dysfunction before it progresses to evident systemic insulin resistance and full-blown cardiometabolic disease is a current challenge. Here, we add to the understanding of immunometabolic heterogeneity within obesity by comprehensively evaluating the association of 15 circulating adipokines with measures of adiposity, cardiometabolic health, and gut microbiota in 116 adults with varying risk of cardiometabolic disease.

### 3.1. Heterogeneity Within Obesity

It has been suggested that not all individuals with overweight or obesity develop metabolic abnormalities [9,10,29]. One significant observation in our study was the contrasting metabolic profiles between subjects with obesity who were cardiometabolically healthy or unhealthy, supporting the concept of metabolic heterogeneity within obesity. Despite their excessive fat accumulation (increased waist circumference, weight, and body fat), individuals with obesity who are cardiometabolically healthy exhibited none or mild metabolic abnormalities compared to individuals with obesity who are cardiometabolically abnormal. This was evidenced in parameters like blood pressure, blood lipids, glycemic profile, and gut microbiota. While both groups displayed signs of low-grade inflammation (e.g., elevated hs-CRP) and altered adipokine levels (higher leptin, IL-18, and chemerin; lower adiponectin), indicative of adipose tissue dysfunction, the alterations in the circulating levels of these inflammatory markers were less pronounced in the obesity healthy phenotype, suggesting a potential threshold effect, whereby the degree of adiposity, the distribution and size of fat cells, the anatomical distribution of fat (e.g., abdominal vs. gluteofemoral), and associated metabolic abnormalities would determine the severity of health outcomes.

In spite of the differences between cardiometabolically healthy and abnormal obesity, there is no consensus about the former phenotype. This condition is proposed to be transitional, as it involves early signs of adipose tissue and metabolic abnormalities that have not yet progressed to full-blown metabolic disease [9]. In the interim, the adipose tissue can expand healthily to accommodate increased fat storage without triggering inflammation [30]. However, when the adipose tissue expands in an unhealthy way, with larger adipocytes, it secretes more proinflammatory factors like TNF-α, IL-6, IL-8, visfatin, leptin, and MCP-1 in response to the hypoxia developed as the tissue mass grows [31,32,33]. Larger adipocytes also exhibit reduced insulin sensitivity, increased lipolysis, and the release of free fatty acids into the bloodstream [34]. Some adipokines can stimulate the liver to produce C-reactive protein [35]. Here, we found higher levels of hs-CRP in individuals with abdominal obesity and in those with adequate waist but metabolically abnormal, at different extents. Elevated hs-CRP impairs the insulin signaling pathway that promotes glucose translocation [36] by phosphorylating the insulin receptor substrate-1, thereby promoting insulin resistance [37]. In essence, our results highlight the connection between low-grade inflammation, adipose tissue dysfunction, and metabolic abnormalities. Consequently, the ability of the adipose tissue to expand healthily is crucial for preventing metabolic complications. As such, inflammation is not solely a result of excess fat but also a consequence of adipose tissue dysfunction, explaining the coexistence of individuals with obesity who are metabolically healthy and with healthy weight who are metabolically abnormal [30], this highlights the importance of viewing adipose tissue health as a dynamic and evolving process, rather than a static condition; recognizing its plasticity and capacity for adaptation is essential for devising effective strategies to prevent and manage obesity-related metabolic complications.

### 3.2. Gut Microbiota and Metabolic Health

Inflammasomes and their product IL-18 play important roles in gut microbiota monitoring and homeostasis [38]. Gut microbiota dysbiosis can lead to increased production of IL-18 by gut macrophages to induce the production of antimicrobial peptides [39], but also contribute to a more inflamed gut state and can further exacerbate dysbiosis [40]. Along with elevated levels of hs-CRP, we observed a significant borderline increase in IL-18 and a six-fold increase in potentially harmful gut bacteria (CAG-Pathogen) in individuals with cardiometabolically abnormal obesity. The positive association between IL-18 and the abundance of pathogenic microbes observed in our study, and its negative association with beneficial microbes (fermentative bacteria grouped in the CAG-Ruminococcaceae), supports the idea that IL-18 constitutes an important link between adiposity, low-grade inflammation, gut dysbiosis, and cardiometabolic risk. On the other hand, TMA and TMA-O emerged as crucial players in the diet–gut microbiota–cardiometabolic health axis [41]. TMAO exacerbates glucose intolerance and beta-cell dysfunction, likely through oxidative stress and inflammation. While some gut bacteria can increase TMAO production, others may mitigate its effects. Despite our careful examination of the gut microbiota, we did not observe differences in the levels of TMA or TMA-O between cardiometabolically healthy and abnormal subjects.

Beyond cytokine-mediated effects, microbiome-derived metabolites play a crucial role in systemic metabolic homeostasis and the development of chronic diseases, including type 2 diabetes and cardiovascular diseases [42]. Short-chain fatty acids (SCFAs), such as propionate and acetate, significantly influence metabolic regulation by modulating hepatic gluconeogenesis, lipogenesis, and adipocyte differentiation. Similarly, branched-chain amino acids (BCAAs) have been shown to disrupt insulin signaling by activating the mTOR kinase complex. This activation promotes protein synthesis, stimulates S6K1 (a kinase essential for cell growth), and enhances the phosphorylation of insulin receptor substrate 1, ultimately impairing insulin sensitivity. However, in our cohort, we previously demonstrated that plasma levels of SCFAs and BCAAs were similar between individuals with cardiometabolically healthy and abnormal statuses [43]. These findings reinforce that other factors, such as microbial composition or host response, may play a more significant role in metabolic differences observed between these groups.

### 3.3. Adipokines as Biomarkers

The dysregulation of adipokines makes them valuable biomarkers for assessing the health status of the adipose tissue and predicting cardiometabolic risk. Previous studies found that elevated levels of leptin, hs-CRP, IL-6, and reduced levels of adiponectin are linked to obesity, but not necessarily diabetes [44,45,46]. This suggests that their alteration may be attributed to greater adipose tissue mass rather than metabolic dysfunction.

Chemerin has been associated with obesity and metabolic syndrome in adults and children [47,48]. Multiple reports show that chemerin levels increase in both rodents and humans with obesity [23], exhibiting contradictory roles in adipogenesis, inflammation, glucose metabolism, and angiogenesis. Cross-sectional studies also indicate that this adipokine is crucial in the pathogenesis of the metabolic syndrome and suggest that chemerin, IL-18, and adiponectin may reciprocally participate in its development [23,48,49]. In normal glucose-tolerant individuals, chemerin levels were linked to BMI, triglycerides, and blood pressure [50,51]. Interestingly, in patients with type-2 diabetes, chemerin was associated with adipose tissue dysfunction, independent of body weight [46].

Several studies also associate IL-18 with metabolic dysfunction. For instance, mice lacking IL-18 or the IL-18 receptor (IL-18R) develop obesity and insulin resistance [24]. Administering recombinant IL-18 reduces appetite, feed efficiency, and weight regain in food-deprived animals [52,53]. Higher levels of IL-18 have been observed in men with metabolic syndrome [54,55] and associated with vascular events in patients with angina or congenital heart disease. In these cases, the NLRP3 inflammasome-dependent activation of IL-18, but not IL-1β, triggers cardiac inflammation and fibrosis [56]. However, in obesity and insulin resistance, IL-18 appears to counteract cardiometabolic dysfunction via the NLRP1 inflammasome [21,24]. Increased IL-18 signaling is also observed within the adipose tissue. Compared to individuals with adequate weight, those with overweight/obesity but without diabetes and individuals with type-2 diabetes show increased expression of the IL-18R and IL-18 mRNA/protein. This suggests that the adipose tissue IL-18R/IL-18 expression is increased in obesity with insulin resistance [57], probably due to the action of leptin, which increases IL-18 secretion in human monocytes via modulation of the caspase-1 inflammasome, without affecting the IL-18 and IL-1β mRNA levels, or IL-1β secretion.

Adiponectin sensitizes cells to insulin, stimulates fat oxidation in the liver and muscle, and inhibits hepatic glucose production. Adiponectin levels typically decrease, whereas leptin levels increase in obesity, insulin resistance, type-2 diabetes, and metabolic syndrome, ultimately leading to increased cardiovascular risk [58,59]. Paradoxically, some individuals with cardiometabolic abnormalities exhibit high levels of both adipokines. [60]. Therefore, the adiponectin/leptin ratio has emerged as a promising biomarker for assessing the cardiometabolic risk associated with adipose tissue dysfunction. Unlike separate adiponectin and leptin levels, which can be influenced by factors such as sex, age, and acute stressors, this ratio is often linked to increased insulin resistance, inflammation, and the development of cardiometabolic complications, such as type-2 diabetes and cardiovascular diseases [61].

We acknowledge the limitations of this study. First, it was cross-sectional, thus, we cannot determine whether the identified patterns on adipokines cause or merely reflect cardiometabolic abnormalities. Second, the sample size of this study was relatively small, which may limit the generalizability of our results. Nevertheless, it provides an opportunity to generate hypotheses for future research with larger cohorts exploring the cause–effect relationships between adipokines, gut microbiota, inflammation, and the development of metabolic abnormalities within obesity. Furthermore, research on adipokines and inflammation as early markers of cardiometabolic risk in individuals with obesity is warranted.

## 4. Materials and Methods

### 4.1. Study Population

This study used a convenience sample of community-dwelling individuals who had been previously identified as having distinct gut microbiota dominated by a single consortium of co-abundant microorganisms (CAGs) and varying cardiometabolic disease risk [25,26]. CAGs were defined by calculating Spearman’s correlation coefficients between all OTUs found after 16S rRNA gene sequencing and by applying hierarchical clustering with Ward’s linkage. OTUs with the highest median abundances served to name each CAG. The study population was described in detail elsewhere [26]. Briefly, between July and November 2014, 441 men and women aged 18–62 years with a BMI ≥ 18.5 kg/m^2^ were enrolled, residing in the Colombian cities of Bogotá, Medellin, Cali, Barranquilla, and Bucaramanga, the country’s largest urban centers. All participants were insured by the health insurance provider EPS SURA. Underweight individuals (BMI < 18.5 kg/m^2^), pregnant women, individuals who had consumed antibiotics or antiparasitics within the three months prior to enrollment, individuals diagnosed with any of the following diseases: Alzheimer’s disease, Parkinson’s disease, or any other neurodegenerative disease, current or recent cancer (within the past year), gastrointestinal diseases (Crohn’s disease, ulcerative colitis, short bowel syndrome, diverticulosis, or celiac disease) were excluded.

### 4.2. Subjects

Data of 116 individuals of the original cohort were retrieved from the studied population, maintaining similar proportions of sex (60 males and 56 females), age range (55 individuals 18–40 years, 61 individuals 41–62 years), dominant gut microbiota (CAG-Prevotella: 22 individuals, CAG-Lachnospiraceae: 27, CAG-Pathogen: 24, CAG-Akkermansia: 23, CAG-Ruminococcaceae: 20), BMI (adequate weight: 35 individuals, overweight: 49, obesity: 32), and abdominal obesity (waist circumference >86 cm for women: 29 individuals; >89 cm for men: 46) [62]. Demographics, weight, height, waist circumference, body fat percentage, systolic (SBP) and diastolic (DBP) blood pressures, and blood chemistry parameters (HDL cholesterol, LDL cholesterol, very low-density lipoprotein (VLDL) cholesterol, oxidized LDL, triglycerides, fasting insulin, fasting glucose, leptin, and total adiponectin) were retrieved from published records [25,26,63]. Subjects were classified to exhibit a cardiometabolically healthy or abnormal health status based on criteria established by Tomiyama et al. [64]. That is, individuals with cardiometabolically abnormal traits were those having two or more of the following metabolic abnormalities: (1) SBP/DBP ≥ 130/85 mm Hg or consumption of antihypertensive medication; (2) fasting blood sugar ≥ 100 mg/dL or consumption of antidiabetic medication; (3) HOMA-IR > 3; (4) hs-CRP > 3 mg/L; (5) serum triglycerides ≥ 150 mg/dL; and (6) HDL < 40 mg/dL (men) or <50 mg/dL (women) or consumption of lipid-lowering medication.

### 4.3. Adipokines Determination

Adipokine levels were measured in plasma using a customized human magnetic premixed multiplex assay kit (R&D Systems, Minneapolis, MN, USA), and Magpix (Luminex Corp, Austin, TX, USA) was used to quantify the plasma levels of CCL-2/MCP-1, IL-1β/IL-1F2, IL-8/CXCL8, IL-33, PBEF/visfatin, Serpin A12/vaspin, chemerin, CXCL5/ENA-78, IL-6, IL-18/IL-1F4, lipocalin-2/NGAL, resistin, and TNF-α. The manufacturer’s recommendations were followed for analysis. The Luminex Xponent software v4.3 was used for data analysis. The limits of detection in pg/mL for each molecule were as follows: MCP-1: 9.9, IL-1β: 0.8, IL-8: 1.8, IL-33: 1.8, visfatin: 2243, vaspin: 2.85, chemerin: 69.0, CXCL5: 8.2, IL-6: 1.7, IL-18: 1.93, lipocalin-2: 29.2, resistin: 3.0, and TNF-α: 1.2, respectively.

### 4.4. TMA and TMA-O Sample Preparation and Analysis

Plasma samples (210 μL) were subjected to protein precipitation using 70 μL of trichloroacetic acid (30% in water), vortexed, left to stand for 5 min, and then centrifuged. A total of 150 µL of the supernatant was taken and 100 µL of water was added, vortexed, and centrifuged if turbid. Following centrifugation, the supernatant was subjected to liquid–liquid extraction with chloroform (200 μL). Two aliquots of 100 µL each were recovered from the aqueous layer of the plasma. For TMA analysis, a 100 μL aliquot of the aqueous phase was alkalinized with 20 μL ammonium hydroxide (10% *v*/*v* in acetonitrile), mixed manually and transferred into a vial insert. To quantify TMA-O, 10 μL of 30% titanium(III) chloride was added to the 100 μL aqueous phase and incubated 1 h at 40 °C to reduce TMA-O to TMA and then neutralized with 10 μL of 10% ammonium hydroxide in acetonitrile, mixed manually, and transferred into a vial insert. One μL of both samples was injected into the GC/MS.

The GC/MS analysis was conducted using an Agilent 7890 GC system (Wilmington, DE, USA) equipped with a 5975A MS detector and a 7683B automatic liquid sampler CTC Combipal 3, with a capillary column: DB-Wax MSi (30 m × 0.25 mm; film thickness: 0.25 μm; Agilent J&W Scientific, Santa Clara, CA, USA). The inlet and interface temperatures were set to 220 °C and 260 °C, respectively. Electron impact ionization energy was set to 70 eV, and data acquisition was performed in single ion monitoring mode. The oven temperature was programmed to start at 40 °C for 5 min, ramping at 5 °C/min to 50 °C, and then at 30 °C/min to 260 °C for 5 min, using helium as a carrier gas and operating in splitless injection mode. Quantification was achieved using isotope-labeled internal standards (d_9_-TMA and d_9_–TMA-O). Calibration curves were generated using a series of standard solutions.

### 4.5. Statistical Analysis

Variables were categorized by cardiometabolic health status (healthy, abnormal), waist circumference (adequate waist, abdominal obesity), BMI (adequate weight, overweight, obesity), and gut microbiota composition (dominant CAG). Data are reported as the mean ± SD and 95% confidence intervals (CIs). The Shapiro–Wilks test was used to determine the normality of the continuous variables in each group. To visually compare variables between sub-groups, Z-scores were calculated and plotted. Next, multiple linear model analysis was performed on data; a constant was added to variables with zero values (less than 5% of all variables evaluated) to avoid infinite values during logarithmic transformation to fit a normal distribution. Analyses were adjusted for potential confounding factors such as age (18–40 and 41–62 years), sex (male, female), and city (Bogota, Medellin, Cali, Barranquilla, and Bucaramanga). All analyses were performed with the R-statistical open-source software version 4.0.5 (R Core Team, 2019).

## 5. Conclusions

Our study adds to the understanding of immunometabolic heterogeneity within obesity by offering a comprehensive evaluation of adipokines, adiposity, cardiometabolic health, and gut microbiota. We provide evidence of the existence of the cardiometabolically healthy obesity phenotype and highlight the potential of adipokines such as leptin, adiponectin, IL-18, and chemerin; the inflammatory marker hs-CRP; and gut dysbiosis, as indicators of the evolution from the cardiometabolically healthy to the cardiometabolically abnormal phenotype within this population. Future research should prioritize uncovering the mechanisms of these associations and explore the clinical potential of these markers for early detection and intervention.

## Figures and Tables

**Figure 1 ijms-26-01176-f001:**
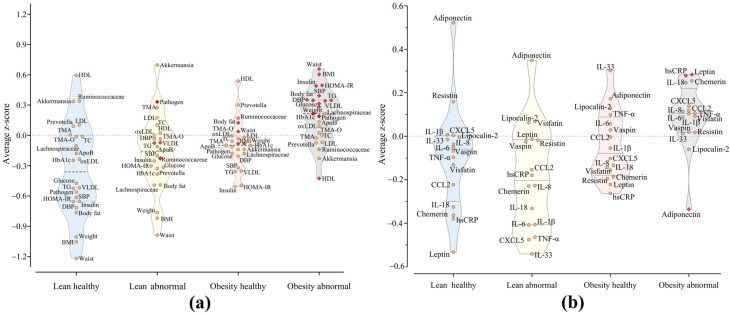
Distinct clinical and adipokine profiles are associated with metabolic health in subjects with overweight/obesity. (**a**) Violin plots of Z-scores for anthropometry, blood chemistry, and gut microbiota of subjects classified by waist circumference and cardiometabolic health status into: lean healthy (light blue), lean abnormal (light yellow), obesity healthy (pink) and obesity abnormal (gray). (**b**) Violin plots of Z-scores comparing the levels of adipokines in the same groups of subjects.

**Figure 2 ijms-26-01176-f002:**
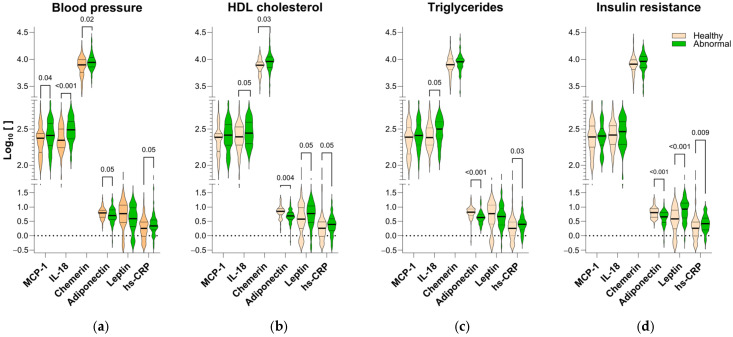
hs-CRP and adiponectin levels are consistently altered in metabolic abnormalities. (**a**) Blood pressure (healthy = 61; abnormal = 55); (**b**) HDL (healthy = 48; abnormal = 68), (**c**) triglycerides (healthy = 71; abnormal = 45), and (**d**) insulin resistance (HOMA-IR, healthy = 74; abnormal = 42). Groups were compared through unpaired *t*-tests with Welch correction. Differences with *q* < 0.05 after adjusting *p*-values for multiple comparisons, controlling the False Discovery Rate (FDR) set at 5% by the two-stage step-up (Benjamini, Krieger, and Yekutieli) were considered a discovery. hs-CRP: high-sensitivity C reactive protein; IL: interleukin; MCP-1: monocyte chemoattractant protein-1.

**Table 1 ijms-26-01176-t001:** Distribution of measured variables in the studied population. Subjects classified according to the four cardiometabolic phenotypes associated with abdominal obesity. Data are presented as the mean ± standard deviation (SD) and 95% confidence intervals (CI). To compare all groups, we employed a multiple linear regression model (MLR) with log-transformed variables. The model was adjusted for age range, sex at birth, smoking, and city of origin. Reported *p*-values indicate the statistical significance of these comparisons. Additionally, statistical significance in post hoc analyses is denoted by * for *p* < 0.05 after correction for multiple comparisons using Dunn’s test and represents the contrasts between the phenotypes lean healthy and obesity (healthy or abnormal), and ^#^ for *p* < 0.05 within the contrast between obesity healthy and obesity abnormal.

	Lean Healthy		Lean Abnormal		Obesity Healthy		Obesity Abnormal		*p*-Value	
	Mean ± SD	CI 95%	Mean ± SD	CI 95%	Mean ± SD	CI 95%	Mean ± SD	CI 95%	Model 1	Model 2
N	26	-	15	-	18	-	57	-	-	-
Age (years)	36.4 ± 11.6	32.0; 41.4	38.2 ± 11.0	31.0; 45.0	43.3 ± 10.9	37.9; 48.7	42.2 ± 11.8	39.1; 45.3	-	-
Female (n)	11	-	9	-	11		34	-	-	-
Male (n)	11	-	4	-	11		25	-	-	-
*Anthropometry*										
BMI (kg/m^2^)	23.07 ± 2.02	22.25; 23.88	24.03 ± 2.45	22.66; 25.37	28.10 ± 2.91 *	26.65; 29.55	30.84 ± 3.96 *	29.79; 31.90	<0.0001	<0.0001
Weight (kg)	59.57 ± 6.18	57.07; 62.07	61.83 ± 9.66	56.48; 67.18	78.22 ± 12.37 *	72.07; 84.37	84.10 ± 12.72 *	80.73; 87.48	<0.0001	<0.0001
Waist circumference (cm)	78.81 ± 4.38	77.04; 80.58	80.74 ± 4.88	78.04; 83.45	96.33 ± 6.19 *	93.26; 99.41	103.05 ± 9.23 *	100.60; 105.50	<0.0001	<0.0001
Body fat (%)	33.85 ± 5.27	31.72; 35.98	35.35 ± 3.80	33.25; 37.45	38.11 ± 3.00 *	36.62; 39.61	39.11 ± 5.38 *	37.68; 40.54	<0.0001	<0.0001
*Blood pressure*										
Systolic (mm Hg)	114.77 ± 9.95	110.75; 118.79	120.80 ± 25.33 *	106.77; 134.83	119.22 ± 19.91	109.32; 129.12	133.77 ± 17.13 *^,#^	129.23; 138.32	<0.0001	<0.0001
Diastolic (mm Hg)	70.31 ± 7.54	67.26; 73.35	77.53 ± 16.63	68.33; 86.74	78.83 ± 11.83	72.95; 84.72	84.65 ± 11.01 *	81.83; 87.57	<0.0001	<0.0001
*Blood lipids*										
HDL (mg/dL)	52.15 ± 9.16	48.46; 55.85	45.07 ± 7.82	40.73; 49.40	48.56 ± 6.12	45.51; 51.60	39.47 ± 11.25 *^,#^	36.49; 42.46	<0.0001	<0.0001
VLDL (mg/dL)	18.74 ± 6.72	16.02; 21.45	28.43 ± 11.04 *	22.31; 34.54	22.40 ± 10.13	17.36; 27.44	37.47 ± 27.62 *^,#^	30.14; 44.80	<0.0001	<0.0001
LDL (mg/dL)	122.50 ± 39.11	106.70; 138.30	127.00 ± 33.65	108.36; 145.64	114.39 ± 23.63	102.64; 126.14	113.30 ± 32.84	104.51; 122.10	0.472	0.362
Total cholesterol (mg/dL)	191.62 ± 44.84	173.50; 209.73	197.27 ± 36.61	176.99; 217.54	181.39 ± 29.86	166.54; 196.24	187.58 ± 39.75	177.03; 198.12	0.686	0.460
Triglycerides (mg/dL)	93.69 ± 33.48	80.17; 107.21	142.40 ± 55.57 *	111.62; 173.18	112.06 ± 50.28	87.05; 137.06	188.65 ± 137.64 *^,#^	152.13; 225.17	<0.0001	<0.0001
oxLDL (U/L)	137.29 ± 54.41	114.83; 159.75	179.96 ± 84.58	133.11; 226.80	178.26 ± 77.21	139.86; 216.66	176.34 ± 131.18	140.18; 212.49	0.465	0.157
ApoB (mg/dL)	90.67 ± 27.49	79.56; 101.77	94.57 ± 15.84	85.79; 103.34	91.39 ± 26.33	78.30; 104.49	108.51 ± 84.26	86.15; 130.87	0.421	0.384
*Glycemic profile*										
HbA1c (%)	5.42 ± 0.30	5.30; 5.54	5.31 ± 0.27	5.16; 5.46	5.41 ± 0.34	5.30; 5.59	5.59 ± 0.50 *	5.46; 5.72	0.075	0.018
Glucose (mg/dL)	83.42 ± 7.06	80.57; 86.27	85.53 ± 9.46	80.29; 90.77	85.00 ± 4.10	82.96; 87.04	92.72 ± 17.24 *	88.14; 97.29	0.005	0.050
Insulin (µU/mL)	7.90 ± 2.41	6.93; 8.88	12.65 ± 5.91 *	9.38; 15.93	9.15 ± 2.85	7.74; 10.57	17.67 ± 10.27 *^,#^	14.95; 20.40	<0.0001	<0.0001
HOMA-B	158.73 ± 89.12	122.74; 194.73	228.27 ± 138.21	151.73; 304.81	153.96 ± 58.47	124.89; 183.04	259.53 ± 177.39 *	212.46; 306.60	<0.0001	<0.0001
HOMA-S	68.17 ± 24.15	58.42; 77.93	49.65 ± 30.20	32.92; 66.37	58.37 ± 22.37	47.25; 69.49	33.85 ± 20.30 *^,#^	29.05; 40.03	<0.0001	<0.0001
HOMA-IR	1.63 ± 0.49	1.43; 1.82	2.68 ± 1.29	1.96; 3.40	1.93 ± 0.63	1.61; 2.24	4.09 ± 2.63 *^,#^	3.39; 4.79	<0.0001	<0.0001
*Gut microbiota and metabolites*										
CAG-Prevotella	0.17 ± 0.22	0.08; 0.26	0.09 ± 0.16	−0.003; 0.18	0.27 ± 0.29	0.13; 0.41	0.15 ± 0.22	0.09; 0.21	0.466	0.578
CAG-Lachnospiraceae	0.20 ± 0.22	0.11; 0.29	0.18 ± 0.17	0.08; 0.27	0.20 ± 0.22	0.09; 0.31	0.27 ± 0.25	0.21; 0.34	0.483	0.603
CAG-Pathogen	0.04 ± 0.06	0.02; 0.06	0.19 ± 0.28	0.04; 0.35	0.15 ± 0.21	0.05; 0.26	0.24 ± 0.30	0.16; 0.32	0.034	0.166
CAG-Akkermansia	0.25 ± 0.29	0.14; 0.37	0.34 ± 0.31	0.17; 0.51	0.14 ± 0.20	0.04;0.24	0.12 ± 0.19 *	0.07; 0.17	0.004	0.043
CAG-Ruminococcaceae	0.17 ± 0.17	0.10; 0.24	0.07 ± 0.08	0.03; 0.12	0.10 ± 0.14	0.03; 0.17	0.09 ± 0.13	0.05; 0.12	0.067	0.109
TMA (µM)	1.84 ± 0.63	1.59; 2.09	2.04 ± 0.59	1.71; 2.37	1.82 ± 0.68	1.48; 2.16	1.84 ± 0.67	1.66; 2.02	0.637	0.438
TMA-O (µM)	3.66 ± 0.55	3.46; 3.91	3.65 ± 0.57	3.34; 3.97	3.60 ± 0.64	3.28; 3.92	3.67 ± 0.52	3.53; 3.81	0.934	0.968
*Adipokines and inflammation markers*										
hs-CRP (mg/L)	1.31 ± 0.90	0.94; 1.68	2.15 ± 1.21	1.48; 2.82	1.97 ± 1.11	1.41; 2.52	4.42 ± 6.15 *^,#^	2.78; 6.05	<0.0001	<0.0001
TNF-α (pg/mL)	13.07 ± 5.60	10.81; 15.34	9.75 ± 4.10	7.48; 12.03	15.06 ± 7.11	11.52; 18.59	14.96 ± 9.56	12.42; 17.49	0.042	0.090
IL-6 (pg/mL)	5.68 ± 4.04	4.05; 7.31	3.75 ± 2.67	2.27; 5.22	6.65 ± 4.85	4.24; 9.06	6.66 ± 5.599	5.07; 8.25	0.105	0.190
IL-33 (pg/mL)	127.45 ± 15.16	121.32; 133.57	118.92 ± 19.43	108.16; 129.68	136.01 ± 23.05	124.55; 147.47	128.67 ± 17.53	124.02; 133.02	0.064	0.143
IL-8 (pg/mL)	33.36 ± 45.22	15.10; 51.63	20.72 ± 25.69	6.50; 34.95	30.33 ± 32.17	14.34; 46.33	46.67 ± 86.54	23.70; 69.63	0.324	0.206
MCP-1 (pg/mL)	229.28 ± 105.71	186.59; 271.98	245.41 ± 134.01	171.28; 319.62	286.75 ± 166.51	203.94; 369.55	284.05 ± 119.69	252.30; 315.81	0.134	0.207
IL-1β (pg/mL)	11.61 ± 1.69	10.92; 12.29	10.65 ± 2.05	9.51; 11.78	11.46 ± 2.25	10.34; 12.58	11.89 ± 2.52	11.22; 12.56	0.310	0.251
Visfatin (ng/mL)	2.68 ± 2.41	1.71; 3.65	3.05 ± 3.36	0.89; 4.61	2.19 ± 1.89	1.25; 3.13	3.19 ± 2.82	2.45; 3.94	0.430	0.565
Resistin (ng/mL)	13.00 ± 5.52	10.76; 15.23	12.42 ± 2.94	10.79; 14.05	12.47 ± 4.72	10.12; 14.82	13.07 ± 6.21	11.43; 14.72	0.993	0.957
Lipocalin-2 (ng/mL)	28.79 ± 4.61	26.92; 30.65	29.00 ± 3.53	27.04; 30.95	29.34 ± 6.17	26.27; 32.41	28.40 ± 4.86	27.16; 29.68	0.913	0.796
CXCL5 (ng/mL)	1.40 ± 0.90	1.03; 1.76	1.32 ± 0.88	0.84; 1.81	1.31 ± 0.87	0.88; 1.74	1.47 ± 1.16	1.17; 1.78	0.916	0.711
Chemerin (ng/mL)	7.34 ± 2.23	6.44; 8.24	8.06 ± 1.84	7.04; 9.08	8.47 ± 2.89	7.04; 9.91	9.59 ± 3.89	8.56; 10.62	0.056	0.058
Vaspin (ng/mL)	2.16 ± 8.98	−0.84; 5.16	2.54 ± 5.60	−0.56; 5.63	4.07 ± 8.41	−0.12; 8.25	3.37 ± 9.40	0.88; 5.87	0.331	0.460
IL-18 (pg/mL)	240.66 ± 102.93	199.09; 282.23	236.95 ± 88.97	187.68; 286.22	289.18 ± 84.11	248.81; 329.54	323.17 ± 131.74	288.21; 358.12	0.008	0.102
Leptin (ng/mL)	4.27 ± 3.46	2.87; 5.67	7.54 ± 6.79	3.79; 11.30	5.21 ± 4.36	3.04; 7.37	8.94 ± 7.61 *	6.92; 10.96	0.004	<0.001
Adiponectin (µg/mL)	8.35 ± 3.66	6.87; 9.83	7.55 ± 4.11	5.27; 9.82	6.19 ± 2.28	5.06; 7.32	5.08 ± 2.73 *	4.35; 5.80	<0.0001	<0.001
Adiponectin/leptin	5.42 ± 5.85	3.06; 7.79	1.74 ± 1.35	0.99; 2.49	2.72 ± 3.04	1.21; 4.23	1.27 ± 1.62 *^,#^	0.84; 1.70	<0.0001	<0.0001

BMI: body mass index; HDL: high-density lipoprotein cholesterol; VLDL: very low-density lipoprotein cholesterol; LDL: low-density lipoprotein cholesterol; oxLDL: oxidized low-density lipoprotein cholesterol; ApoB: apolipoprotein B; TMA: trimethylamine; TMA-O: trimethylamine N-oxide; HbA1c: glycated hemoglobin; HOMA: homeostasis model assessment; CAG co-abundance groups; hs-CRP high-sensitivity C reactive protein; TNF-α: tumor necrosis factor alpha; IL: interleukin; MCP-1: monocyte chemoattractant protein-1; CXCL5: C-X-C motif chemokine ligand 5. Model 1. MLR unadjusted. Model 2 MLR adjusted for potential confounders age range, sex at birth, smoking, and city of origin.

## Data Availability

The original data and code presented in the study are openly available in Github at [https://github.com/vidarium/adipokines] accessed on 20 January 2025.

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
