# Peer review of "Circulating hs-CRP, IL-18, Chemerin, Leptin, and Adiponectin Levels Reflect Cardiometabolic Dysfunction in Adults with Excess Weight"

_ijms, 2025, doi:10.3390/ijms26031176_

Round 1
Reviewer 1 Report
Comments and Suggestions for Authors
The article submitted for publication by Lara-Guzmán et al., titled: "Circulating hs-CRP, IL-18, chemerin, leptin, and adiponectin levels reflect cardiometabolic dysfunction in adults with excess weight" is aiming to investigate an interesting question on the potential of using certain biomarkers (namely circulating cytokines) as an early marker of potential cardiometabolic predictors.
The paper is well written and reasonably structured and organized.
The reviewer would like to offer the following points below for consideration by the authors:
1. Consider including some statistics in terms of cardiovascular disease and diabetes in the introduction section, regarding the prevalence and the burden of disease.
2. The authors state that they correlated body composition with circulating cytokines. Yet they did use waist circumference which although it correlates with abdominal fat cannot really substitute for an actual body composition measurement/assessment as conducted through DXA or BIA. Thus it would be more accurate and advisable instead of using the term body composition express what the authors actually did which is waist circumference measurements.
3. BMI does not have units. It is an index and weight measured/expressed in Kg and height measured/expressed in m are used to calculate (not measure) the index (BMI).
4. Consider specifying clearly the inclusion and exclusion criteria for participants in the study.
5. What was the rationale for the determination of sample size in the study?
6. Did the authors control for (normalize) factors which may impact certain biomarkers such as CRP like smoking, alcohol consumption, being sick with seasonal flu for example etc?
7. Consider discussing directly the relationship between the gut microbiome and diabetes and ensuing CVD regardless of cytokines a brief additional discussion in this regard would benefit the paper. A paper that might help in that is: Sikalidis, A.K.; Maykish, A. The Gut Microbiome and Type 2 Diabetes Mellitus: Discussing A Complex Relationship. Biomedicines 2020, 8, 8. https://doi.org/10.3390/biomedicines8010008.
Author Response
Response to Reviewer 1 Comments
Thank you very much for taking the time to review this manuscript. Please find the detailed responses below and the corresponding revisions/corrections highlighted/in track changes in the re-submitted files.
Point-by-point response to Comments and Suggestions
Comment 1: Comment 1. Consider including some statistics in terms of cardiovascular disease and diabetes in the introduction section, regarding the prevalence and the burden of disease.
Response 1: Thank you for pointing this out. We agree with this comment. Therefore, we have added an additional paragraph to the manuscript in the introduction section on page 1, paragraph 1, lines 33 to 41.
“Cardiovascular disease (CVD) and type 2 diabetes mellitus (T2DM) are major global health challenges, with significant morbidity and mortality worldwide. The economic and social burden of these diseases is substantial, impacting individuals, families, and healthcare systems. An estimated 20.5 million people died from CVDs in 2021, representing one third of all global deaths [1]. Similarly, in 2022 the number of adults living with diabetes worldwide has surpassed 800 million [2] and is a leading cause of kidney failure, lower limb amputations, and blindness. The co-occurrence of obesity, T2DM, and CVD has become a significant public health concern, highlighting the need for a better understanding of the underlying mechanisms”.
Comment 2: The authors state that they correlated body composition with circulating cytokines. Yet they did use waist circumference which although it correlates with abdominal fat cannot really substitute for an actual body composition measurement/assessment as conducted through DXA or BIA. Thus it would be more accurate and advisable instead of using the term body composition to express what the authors actually did, which is waist circumference measurements.
Response 2: The reviewer correctly points out that waist circumference, while a crucial indicator of abdominal adiposity, does not fully capture overall body composition. In this study, we used available anthropometric parameters, including Body Mass Index (BMI), height, and % body fat (assessed using skinfold measurements), in addition to waist circumference, to characterize the participants' body composition.
Waist circumference was used as a key indicator of abdominal obesity, as it is a well-established and clinically relevant measure of central adiposity. Therefore, to be more precise, we have changed body composition for “central obesity” on page 2, paragraph 5, line 89; “higher adiposity” on page 4. paragraph 2, line 129; “healthy status” on page 5 paragraph 1, line 173; “adiposity” on page 5 paragraph 1, line 222; and “adiposity” on page 13, paragraph 2, line 439;
Comment 3. BMI does not have units. It is an index and weight measured/expressed in Kg and height measured/expressed in m are used to calculate (not measure) the index (BMI).
Response 3: We respectfully disagree with the reviewer's statement that BMI does not have units. The Body Mass Index (BMI) is a mathematical ratio that associates the mass (weight) and height of an individual, where mass is expressed in kilograms (kg) and the square of height in meters squared (m²). BMI can be measured well enough, being a quantity derived from height and weight. Therefore, the unit of measure for BMI in the International System of Units is kg/m². We did not change the manuscript on this regard.
We recognize that BMI is an index; however, it is derived from a mathematical ratio involving specific units of measurement. Similar to other ratios, such as density (mass/volume), BMI retains a unit that facilitates comparison and classification. Even when the measured attribute is an abstract concept like health rather than a physical property such as length or mass, the majority of papers published in this journal present BMI with units of kg/m². For reference, here are some recent examples: https://doi.org/10.3390/ijms26020506; https://doi.org/10.3390/ijms26010393; https://doi.org/10.3390/ijms26010040.
Comment 4. Consider specifying clearly the inclusion and exclusion criteria for participants in the study.
Response 4: The reviewer has rightly pointed out the need for a more detailed description of the inclusion and exclusion criteria. While a detailed description of the study population was previously reported in reference [26]. Therefore, for clarity, we have included the following specific inclusion and exclusion criteria in the revised manuscript on page 11, paragraph 3, lines 358 to 367.
“Briefly, between July and November 2014, were enrolled 441 men and women aged 18-62 years with a BMI ≥ 18.5 kg/m² residing in the Colombian cities of Bogotá, Medellin, Cali, Barranquilla, and Bucaramanga, the country's largest urban centers. All participants were insured by the health insurance provider EPS SURA. Underweight individuals (BMI < 18.5 kg/m²), pregnant women, individuals who had consumed antibiotics or antiparasitics within the three months prior to enrollment, individuals diagnosed with any of the following diseases: Alzheimer's disease, Parkinson's disease, or any other neurodegenerative disease, current or recent cancer (within the past year), gastrointestinal diseases (Crohn's disease, ulcerative colitis, short bowel syndrome, diverticulosis, or celiac disease), were excluded”.
Comment 5. What was the rationale for the determination of sample size in the study?
Response 5: The reviewer raises valid points regarding the sample size and selection criteria. This study utilized a non-probabilistic or convenience sample of 116 individuals from a previously characterized cohort of 441 participants as stated on page 11, paragraph 3, in line 352). The primary objective of this study was to investigate the association between specific gut microbiota compositions and adipokine levels in individuals with varying cardiometabolic risk. While a formal power analysis was not conducted due to the exploratory nature of the study, the sample size of 116 individuals was determined considering the specific research objectives and the complexity of the analyses.
Our group have previously demonstrated associations between gut microbiota and lifestyle, obesity and cardiometabolic disease in a separate analysis of the larger cohort 10.1038/s41598-018-29687-x and showed that the gut microbiota of Colombians has a complex multispecies nature and is better described by an enterogradient (i.e., a continuum of abundances of microbial taxa) rather than specific enterotypes. To manage this complexity, the 100 most abundant OTUs were clustered into five co-abundance groups (CAGs). We therefore, to enhance the clarity and interpretability of our findings, selected individuals with distinct and well-defined gut microbiota profiles, specifically those enriched with particular bacterial consortia (CAGs) while minimizing the presence of other dominant consortia by focusing on the extremes of this microbial gradient. We acknowledge that the reduced sample size may limit the generalizability of our findings. However, by focusing on individuals displaying the most extreme profiles, we aimed to maximize the possibilities to detect potential associations between specific gut microbiota compositions and adipokines. We believe that this approach allowed us to examine the impact of these specific microbial signatures on cardiometabolic health and adipokines profiles more effectively.
Comment 6. Did the authors control for (normalize) factors which may impact certain biomarkers such as CRP like smoking, alcohol consumption, being sick with seasonal flu for example etc.?
Response 6: The reviewer raises a valid point regarding potential confounding factors that can influence circulating CRP levels, such as smoking, alcohol consumption, and recent illness.
In our initial analyses, we adjusted for age, sex at birth, and city of origin as main covariates. We chose not to correct for additional potential confounding variables due to the relatively small sample size, as including too many variables in the model could consume degrees of freedom, reduce statistical power, and make the results less reliable. Recent illness was not adjusted for, as it was an exclusion criterion for the study. Regarding alcohol intake, we identified only 4 heavy drinkers among the 116 subjects. This small subgroup size was considered insufficient for meaningful statistical adjustment, as it would likely yield unstable or unreliable estimates.
We conducted further analyses incorporating smoking as a covariate in our multiple linear regression models. These analyses did not reveal any substantial differences in the results. Ultimately, we assessed the independent association between our variables of interest while controlling for the potential confounding effects of age range, sex at birth, city of origin, and smoking. Therefore, Table 1 was accordingly updated to reflect these additional analyses and their findings on pages 6 and 7.
Comment 7. Consider discussing directly the relationship between the gut microbiome and diabetes and ensuing CVD regardless of cytokines a brief additional discussion in this regard would benefit the paper. A paper that might help in that is: Sikalidis, A.K.; Maykish, A. The Gut Microbiome and Type 2 Diabetes Mellitus: Discussing A Complex Relationship. Biomedicines 2020, 8, 8. https://doi.org/10.3390/biomedicines8010008.
Response 7: We appreciate the recommendation to expand the discussion by directly addressing the relationship between the gut microbiome, diabetes, and cardiovascular disease (CVD), independent of cytokine interactions. The interplay between gut microbiota dysbiosis and the onset and progression of type 2 diabetes mellitus (T2DM) is indeed complex and highly relevant. Alterations in microbial diversity and functionality have been implicated in impaired glucose metabolism, insulin resistance, and systemic inflammation, which are key contributors to diabetes and its complications, including CVD. The suggested reference, Sikalidis and Maykish (2020), provides insights into some mechanisms, particularly highlighting how microbiota metabolites influence metabolic and cardiovascular health independently of inflammation and cytokines as effector molecules. Short chain fatty acids propionate and acetate can modulate gluconeogenesis and lipogenesis in the liver and adipocyte differentiation, while Branched-chain amino acids BCAAs have been shown to interfere with insulin signaling by stimulating mTOR, a kinase complex that plays an important role in protein synthesis, S6K1, a kinase important for cell growth, and phosphorylation of insulin receptor substrate 1. We revised the manuscript and included a brief discussion on page 10, paragraph 2, in lines 284-296.
"Beyond cytokine-mediated effects, microbiome-derived metabolites play a crucial role in systemic metabolic homeostasis and the development of chronic diseases, in-cluding type 2 diabetes and cardiovascular diseases [42]. Short-chain fatty acids (SCFAs), such as propionate and acetate, significantly influence metabolic regulation by modulating hepatic gluconeogenesis, lipogenesis, and adipocyte differentiation. Similarly, branched-chain amino acids (BCAAs) have been shown to disrupt insulin signaling by activating the mTOR kinase complex. This activation promotes protein synthesis, stimulates S6K1 (a kinase essential for cell growth), and enhances the phosphorylation of insulin receptor substrate 1, ultimately impairing insulin sensitivity. However, in our cohort, we previously demonstrated that plasma levels of SCFAs and BCAAs were similar between individuals with cardiometabolically healthy and abnormal statuses [43]. These findings reinforce that other factors, such as microbial composition or host response, may play a more significant role in metabolic differences observed between these groups."
Reviewer 2 Report
Comments and Suggestions for Authors
In the present study authors compared the circulating levels of selected adipokines and gut microbiota composition between individuals with cardiometabolically healthy and unhealthy obesity. The study was performed in 116 patients with overweight or obesity. The topic and the results of this study are of interest, however, there are also important concerns to be addressed.
1. The Abstract should be re-written since the results are present not clearly. It is stated that the study was performed in overweight/obese patients and then in the results section authors describe the difference between those with abdominal obesity and those with normal waist circumference. It should be specified if you compare patients with abdominal obesity to those with non-abdominal obesity or metabolically unhealthy and unhealthy obesity. Then, in lines 22-23 groups with adequate waist or obesity cardiometabolically healthy are compared.
2. Section 4.2: why only 116 subjects were included from the original population? Was the required number of patients calculated before the study? If the patients were selected according to gut microbiota composition, the method used to measure this composition should be described.
3. Line 347, the criteria of metabolic health status should be described in the present manuscript. Referring to prior studies only is not appropriate.
4. It should be specified whether total or HMW adiponectin was measured.
5. Line 392: did normal waist group include only overweight-obese patients or also normal weight patients?
6. Statistical analysis, were all data demonstrated as mean +/- SD regardless of distribution? Or were all data normally distributed?
7. The patients were categorized in the binary manner simply to metabolically healthy and unhealthy groups. However, the former could include those with 0 or 1 metabolic abnormalities and the latter could have 2,3, 4, etc. metabolic abnormalities making both group heterogeneous. The number of cardiometabolic abnormalities could affect the level of adipo- and cytokines.
8. According to the Results, there were actually 4 groups of patients with 2 independent variables (abdominal obesity and metabolic health status). Consequently, 2-factor ANOVA would be the most appropriate method of data analysis because this method takes into consideration the interactions about the independent factors.
9. Was the analysis of data presented in Table 1 adjusted for the potential confounding variables?
Author Response
Thank you very much for taking the time to review this manuscript. Please find the detailed responses below and the corresponding revisions/corrections highlighted/in track changes in the re-submitted files.
Point-by-point response to Comments and Suggestions
Comment 1. The Abstract should be re-written since the results are not presented clearly. It is stated that the study was performed in overweight/obese patients and then in the results section authors describe the difference between those with abdominal obesity and those with normal waist circumference. It should be specified if you compare patients with abdominal obesity to those with non-abdominal obesity or metabolically unhealthy and unhealthy obesity. Then, in lines 22-23 groups with adequate waist or obesity cardiometabolically healthy are compared.
Response 1: We appreciate the reviewer's feedback regarding the clarity of the abstract. In response to this comment, we have revised the abstract to more clearly reflect the study design and findings. The study included 116 individuals categorized into four groups based on central obesity and cardiometabolic health status: normal waist, cardiometabolically healthy, normal waist, cardiometabolically abnormal, abdominal obesity, cardiometabolically healthy, abdominal obesity, cardiometabolically abnormal. The revised abstract in page 1, lines 13 to 29 now more accurately reflects the study design and presents the key findings more concisely.
Abstract: “Up to 30% of individuals with obesity, may exhibit normal insulin sensitivity, a favorable lipid profile, and no signs of hypertension. This prompts the exploration of factors distinguishing cardiometabolically healthy individuals from those developing complications. This cross-sectional study included 116 individuals categorized in four groups by combining abdominal obesity and cardiometabolic health statuses. We compared circulating adipokines and gut microbiota composition between these groups. Individuals with abdominal obesity had higher levels of hs-CRP, TNF-α, MCP-1, IL-18, chemerin, and leptin, and a less favorable gut microbiota composition, including higher levels of potentially harmful bacteria (CAG-Pathogen) and lower levels of beneficial bacteria (CAG-Ruminococcaceae and CAG-Akkermansia), compared to those with adequate waist circumference. Those with obesity but cardiometabolically healthy displayed similar adipokine levels and microbiota composition to those with adequate waist. In contrast, individuals with abdominal obesity cardiometabolically abnormal exhibited significantly higher levels of hs-CRP, IL-18, chemerin, and leptin, and lower levels of adiponectin and CAG-Ruminococcaceae compared to those with abdominal obesity cardiometabolically healthy and adequate waist. Additionally, they differed in hs-CRP and adiponectin/leptin ratio from individuals with obesity cardiometabolically healthy. These findings suggest that altered adipokine profiles and gut microbiota may contribute to the development or persistence of cardiometabolic complications in obesity”.
Comment 2. Section 4.2: why only 116 subjects were included from the original population? Was the required number of patients calculated before the study? If the patients were selected according to gut microbiota composition, the method used to measure this composition should be described.
Response: The reviewer raises valid points regarding the sample size and selection criteria. This study utilized a non-probabilistic or convenience sample of 116 individuals from a previously characterized cohort of 441 participants as stated on page 11, paragraph 3, in line 352. The primary objective of this study was to investigate the association between specific gut microbiota compositions and adipokine levels in individuals with varying cardiometabolic risk. While a formal power analysis was not conducted due to the exploratory nature of the study, the sample size of 116 individuals was determined considering the specific research objectives, the complexity of the analyses, and resource limitations.
Our group have previously demonstrated associations between gut microbiota and lifestyle, obesity and cardiometabolic disease in a separate analysis of the larger cohort 10.1038/s41598-018-29687-x and showed that the gut microbiota of Colombians has a complex multispecies nature and is better described by an enterogradient (i.e., a continuum of abundances of microbial taxa). To manage this complexity, the 100 most abundant OTUs were clustered into five co-abundance groups (CAGs). We therefore, to enhance the clarity and interpretability of our findings, selected individuals with distinct and well-defined gut microbiota profiles, specifically those enriched with particular bacterial consortia (CAGs) while minimizing the presence of other dominant consortia by focusing on the extremes of this microbial gradient. We acknowledge that the reduced sample size may limit the generalizability of our findings. However, by focusing on individuals displaying the most extreme profiles, we aimed to maximize the possibilities to detect potential associations between specific gut microbiota compositions and adipokines. We believe that this approach allowed us to examine the impact of these specific microbial signatures on cardiometabolic health and adipokines profiles more effectively
A brief description of the method used to measure gut microbiota composition was added to the main text on page 1, paragraph, lines 355 to 357:
“CAGs were defined by calculating Spearman’s correlation coefficients between all OTUs found after sequencing, and by applying hierarchical clustering with Ward’s linkage. OTUs with the highest median abundances served to name each CAG, as shown previously”.
Comment 3. Line 347, the criteria of metabolic health status should be described in the present manuscript. Referring to prior studies only is not appropriate.
Response 3: We thank the reviewer for this valuable comment. We understand the importance of clearly defining the criteria used to assess metabolic health status within the context of this study. We explicitly described the criteria used to define metabolic health status on page 12, paragraph 1, in lines 382 to 387.
“individuals with cardiometabolically abnormal traits were those having two or more of the following metabolic abnormalities: 1) SBP/DBP ≥130/85 mm Hg or consumption of antihypertensive medication; 2) fasting blood sugar ≥100 mg/dL or consumption of antidiabetic medication; 3) HOMA-IR >3; 4) hs-CRP >3 mg/L; 5) serum triglycerides ≥150 mg/dL; and 6) HDL <40 mg/dL (men) or < 50 mg/dL (women) or consumption of lipid-lowering medication.”
Comment 4. It should be specified whether total or HMW adiponectin was measured.
Response 4: The reviewer is correct. We measured total adiponectin levels. Therefore, the manuscript was updated the word “total adiponectin” has been added on page 12 paragraph 1, in line 379 for clarity.
Comment 5. Line 392: did the normal waist group include only overweight-obese patients or also normal weight patients?
Response 5: The reviewer raises an important point. To clarify, the "normal waist" group in this study included individuals classified as having normal weight as well as those who were overweight by BMI. This accounts for the fact that excess fat can be stored in various regions, including the abdominal and gluteofemoral areas. Some individuals (especially women) can accumulate fat in the gluteofemoral region rather than the abdomen, allowing them to maintain a normal waist circumference. However, gluteofemoral fat, often referred to as lower body fat, is increasingly recognized for its protective role in metabolic health. This fat distribution is associated with a lower risk of cardiovascular and metabolic diseases, in contrast to the elevated risks linked to abdominal fat.
We introduced this clarification on page 3, paragraph 1, in lines 95 to 99:
“It is important to note that individuals with an adequate waist circumference may include those classified as having normal weight as well as individuals who are overweight based on BMI. This distinction acknowledges that excess fat can be stored in various regions of the body, such as the abdominal and gluteofemoral areas, which may not always correspond directly with BMI classifications”.
Comment 6. Statistical analysis, were all data demonstrated as mean +/- SD regardless of distribution? Or was all data normally distributed?
Response: The reviewer raises a valid point regarding data distribution. Data are presented as mean ± SD and 95% confidence intervals, which are appropriate for describing the central tendency and variability of the data. However, it is crucial to acknowledge that not all variables exhibited perfect normality. We used the Shapiro-Wilk test to assess the normality of continuous variables within each group. For variables that deviated significantly from normality, a logarithmic transformation was applied to improve normality. To ensure the assumptions of the multiple linear regression model were met, variables with zero values (less than 5% of all variables evaluated) were adjusted by adding a small constant before logarithmic transformation. This transformation was necessary for model stability and to avoid issues with infinite values. This approach was chosen to ensure the robustness of the multiple linear regression model, which assumes normally distributed residuals.
We acknowledge that presenting transformed data may not be ideal for all readers. To enhance transparency, we have included the code for statistical analysis as supplementary material for those interested in examining the data in more detail. This is specified in the Data availability statement on page 13, paragraph 8, in line 471.
Comment 7. The patients were categorized in the binary manner simply to metabolically healthy and unhealthy groups. However, the former could include those with 0 or 1 metabolic abnormalities and the latter could have 2,3, 4, etc. metabolic abnormalities making both groups heterogeneous. The number of cardiometabolic abnormalities could affect the level of adipo- and cytokines.
Response 7: The reviewer rightly points out the heterogeneity within our defined 'metabolically healthy' and 'unhealthy' groups. While acknowledging this limitation, we opted for a binary categorization for several reasons. Firstly, this approach aligns with current clinical practice, where individuals are often broadly classified as metabolically healthy or unhealthy. Secondly, a more granular categorization based on the number of metabolic abnormalities would have resulted in very small subgroups within our sample size of 116 subjects (12 possible categories), severely limiting the statistical power and robustness of our analyses.
Comment 8. According to the Results, there were actually 4 groups of patients with 2 independent variables (abdominal obesity and metabolic health status). Consequently, 2-factor ANOVA would be the most appropriate method of data analysis because this method takes into consideration the interactions about the independent factors.
Response 8: The reviewer correctly points out that two-factor ANOVA is a suitable statistical method for examining the main effects of abdominal obesity and metabolic health status, as well as their interaction.
However, in our study, we chose multiple linear regression for several reasons: Multiple linear regression allows for the inclusion and adjustment of potential confounders such as age, sex, and other relevant covariates, which can significantly impact the observed relationships. Two-factor ANOVA, in its basic form, does not directly account for these confounding variables. Multiple linear regression provides greater flexibility in modeling complex relationships between variables, including potential non-linear associations https://doi.org/10.1016/0165-1781(88)90022-4.
We believe that multiple linear regression offered a more robust and comprehensive approach to analyzing our data while controlling for potential confounding factors and providing a deeper understanding of the relationships between our variables of interest.
Comment 9. Was the analysis of data presented in Table 1 adjusted for the potential confounding variables?
Response 9: Yes, the analysis Table 1 has accounted for potential confounders to provide a more accurate and unbiased assessment of the associations between the variables. The provided text specifies this on page 6, in lines 190 and 191
"To compare all groups, we employed a multiple linear regression model (MLR) with log-transformed variables. The model was adjusted for age range, sex at birth, smoking and city of origin”.
Round 2
Reviewer 1 Report
Comments and Suggestions for Authors
The authors have reasonably addressed reviewer's comments.